# Gli1^+^ Progenitors Mediate Glucocorticoid-Induced Osteoporosis In Vivo

**DOI:** 10.3390/ijms25084371

**Published:** 2024-04-16

**Authors:** Puying Yang, Fangyuan Shen, Chengjia You, Feng Lou, Yu Shi

**Affiliations:** 1State Key Laboratory of Oral Diseases and National Clinical Research Center for Oral Diseases, West China Hospital of Stomatology, Sichuan University, Chengdu 610041, China; yangpuying@stu.scu.edu.cn (P.Y.); sfy@stu.scu.edu.cn (F.S.); chengjiayou@stu.scu.edu.cn (C.Y.); loufeng@scu.edu.cn (F.L.); 2Department of Endodontics, West China Hospital of Stomatology, Sichuan University, Chengdu 610041, China

**Keywords:** Gli1^+^ metaphyseal mesenchymal progenitors (MMPs), methylprednisolone, teriparatide, bone mass, glucose uptake

## Abstract

For a wide range of chronic autoimmune and inflammatory diseases in both adults and children, synthetic glucocorticoids (GCs) are one of the most effective treatments. However, besides other adverse effects, GCs inhibit bone mass at multiple levels, and at different ages, especially in puberty. Although extensive studies have investigated the mechanism of GC-induced osteoporosis, their target cell populations still be obscure. Here, our data show that the osteoblast subpopulation among Gli1^+^ metaphyseal mesenchymal progenitors (MMPs) is responsive to GCs as indicated by lineage tracing and single-cell RNA sequencing experiments. Furthermore, the proliferation and differentiation of Gli1^+^ MMPs are both decreased, which may be because GCs impair the oxidative phosphorylation(OXPHOS) and aerobic glycolysis of Gli1^+^ MMPs. Teriparatide, as one of the potential treatments for GCs in bone mass, is sought to increase bone volume by increasing the proliferation and differentiation of Gli1^+^ MMPs in vivo. Notably, our data demonstrate teriparatide ameliorates GC-caused bone defects by targeting Gli1^+^ MMPs. Thus, Gli1^+^ MMPs will be the potential mesenchymal progenitors in response to diverse pharmaceutical administrations in regulating bone formation.

## 1. Introduction

Synthetic glucocorticoids (GCs) are immensely effective in the treatment of inflammatory diseases; however, their therapeutic use is restricted by numerous adverse side effects, such as the induction of osteoporosis. Glucocorticoid-induced osteoporosis (GIO) is the most common secondary cause of osteoporosis and results in fractures. GIO not only compromises the quality of life for patients but also imposes significant burdens on families. Additionally, for adolescents, such fractures may impact subsequent growth and development. The detailed mechanisms of GIO have been extensively studied in recent years. A large number of studies have shown that glucocorticoids directly affect osteoblasts, osteoclasts, and osteocytes, interfering with the proliferation, differentiation, and apoptosis of cells, resulting in reduced bone mass [1]. GCs inhibit angiogenesis in growing bone by inducing endothelial cell senescence [2]. Further, besides inducing bone resorption by targeting osteoclasts [3], excess GCs also inhibit osteoblastogenesis while at the same time inducing osteoblast apoptosis, leading to a rapid and profound decrease in bone formation [4,5,6]. Importantly, excess GCs inhibit the proliferation of mesenchymal stem cells (MSCs) in vivo and direct the MSCs’ differentiation commitment toward adipogenesis at the expense of osteogenesis by inducing adipogenic transcription factors such as PPARγ and C/EBPα or by inhibiting osteoblastogenic master regulators such as Runx2 [7,8,9,10]. Although the mechanism of GCs regulating MSCs’ fate is determined, it is still unclear which population of MSCs mediate the detrimental function of GCs on bone mass in vivo. Postnatal Gli1^+^ metaphyseal mesenchymal progenitors (MMPs) have previously been reported as a major source of osteoblasts in the chondro-osseous junction region in postnatally growing mice [11]. Single-cell RNA sequencing (scRNA-seq) experiments reveal Gli1^+^ MMPs possess high heterogeneity and four subpopulations are identified including chondrocyte-like osteoprogenitor (COP), marrow adipogenic lineage progenitors (MALPs), pre-osteoblasts (Pre-OB), and osteoblasts (OB). Based on the lineage tracing experiments and trajectory analysis, the COP cluster is considered the osteoblast progenitor, which is situated on the top hierarchy of Gli1^+^ populations [12]. These data indicate that Gli1^+^ MMPs are critically involved in physiological bone homeostasis. Hitherto, Gli1^+^ MMPs’ potential responses to GCs are unknown. Fracture risk assessment for all patients with long-term use of GCs is required. Pharmacological management is in high demand for patients who have increased fracture risk. Among those treatment options, teriparatide should be the chosen option, although the detailed mechanism is not fully understood [6,13,14]. Teriparatide is one of the most widely used bone-forming drugs in the world to treat osteoporosis. Teriparatide is a recombinant peptide corresponding to amino acids 1–34 of human parathyroid hormone [15]. Emerging evidence suggests that teriparatide promotes bone formation by regulating MSCs at multiple levels, including increasing osteoblast activity, promoting osteogenesis, and activating quiescent cells on the bone surface [16,17]. Recently, teriparatide has been also reported to increase both the cell proliferation and differentiation of Sox9^+^ MSCs in mice [18]. Most importantly, our work also demonstrates COP function as a teriparatide target and mediates its anabolic function on bone formation [12]. It is intriguing to know whether teriparatide ameliorates GCs’ adverse effect on bone mass due to manipulation of the Gli1^+^ MMPs in vivo.

Here, our data demonstrate that both the osteogenic differentiation and proliferation of Gli1^+^ MMPs are reduced by daily administration of methylprednisolone (MPS), a common synthetic GC. Both the oxygen consumption rate (OCR) and extracellular acidification rate (ECAR) of Gli1^+^ MMPs are impaired in response to MPS. Unlike COP being responsive to teriparatide, the osteoblastic subpopulation (OB) of Gli1^+^ MMPs is the main target of MPS indicated by scRNA-seq and immunostaining. Importantly, teriparatide significantly alleviates the reduction in bone mass caused by MPS injection which also rescues the proliferation and osteogenesis of Gli1^+^ MMPs. These data suggest teriparatide-induced COP might give rise to the OB subpopulation of Gli1^+^ MMPs and increase bone mass consequently.

## 2. Results

### 2.1. Daily Injection of Methylprednisolone (MPS) Disrupts Bone Mass in Mice

To establish the glucocorticoid-induced osteoporosis model, C57BL/6J mice at the age of one month were injected with methylprednisolone (MPS) or vehicle (Veh) for consecutive 28 days before harvest. Three-dimensional reconstructions based on μCT analysis were used to visualize the established osteoporotic model (Figure 1A,F). Quantifying distal femoral trabeculae demonstrated that MPS treatment decreased the bone volume over tissue volume (BV/TV) and trabecular bone thickness (Tb. Th). However, the trabecular number (Tb. N) and trabecular spacing (Tb. Sp) showed no statistically significant difference (Figure 1B–E). Meanwhile, the quantification of cortical bone also confirmed that MPS reduced the cortical thickness (Ct. Th) and cortical area (Ct. Ar) without reducing the total area (Tt. Ar) (Figure 1G–I). The H&E staining displayed a reduction in trabeculae and a thinner cortical bone in response to MPS (Figure 1J,K). Double labeling exhibited MPS led to less new bone formation indicated by dynamic histomorphometry (Figure 1L,M). Consistently, the quantification of mineral apposition rate (MAR) by MPS on the femoral surface was less than that in response to vehicle injection (Figure 1N). Therefore, consistent with the previous finding, MPS impaired the bone mass under consecutive treatment.

### 2.2. MPS Administration Decreases the Proliferation and Osteogenic Differentiation of Gli1^+^ Metaphyseal Mesenchymal Progenitors (MMPs)

To determine the short-term effects of MPS, Gli1-CreER^T2^; tdTomato mice were treated with Tamoxifen(TAM) for three days, following the injection of MPS or vehicle for an additional five days. After the last MPS treatment, the mice were administrated EdU (5-ethynyl-2′-deoxyuridine) four hours before harvest for detection of Gli1^+^ MMP proliferation. The immunostaining of endomucin (Emcn) demonstrated a decreased area in the metaphyseal region of vascular endothelial cells in mice treated with MPS (Figure 2A–C), which was consistent with previous studies [2]. The aggrecan-positive cells labeled the chondrocytes of the growth plate and denoted the boundary of the chondro-osseous junction. The Gli1^+^ MMPs labeled by the red fluorescence in the chondro-osseous junction region were much less in response to MPS (Figure 2D,E). The statistical analysis of tdTomato-positive cells showed the same result intuitively (Figure 2F). The immunostaining of EdU and quantification of the double-positive cells beneath the growth plate demonstrated that MPS decreases the proliferation of the Gli1^+^ MMPs (Figure 2G–I). Likewise, the immunostaining of OSX showed a lower percentage of Gli1^+^ MMPs undergoing osteogenesis upon MPS treatment (Figure 2I–L). Thus, MPS reduced trabecular bone mass may be due to decreasing the cell proliferation and differentiation of Gli1^+^ MMPs.

### 2.3. The Osteoblastic Subpopulation Is the Target of MPS during Inducing Osteoporosis

To investigate the influence of the MPS on Gli1^+^ progenitors, we profiled the transcriptomes of Gli1^+^ MMPs in MPS-treated metaphyseal trabecular bone using single-cell RNA sequencing. The one-month-aged Gli1-CreER^T2^; tdTomato mice were treated with three consecutive days of TAM followed by three daily administrations of MPS before sacrifice. The trabecular bone of the metaphysis was obtained by enzymatic digestion after the separation of the growth plate. The tdTomato^+^ (Gli1^+^) cells were selected from the dissociated cells by flow cytometry as described before [12]. Single-cell RNA-seq was performed with the 10× Genomics platform. Based on our previous studies, the sorted Gli1^+^ (tdTomato^+^) MMPs were divided into five major clusters and visualized by a tSNE plot (Figure 3A,B). In terms of compositional ratios, the percentage of OB clusters was significantly reduced in the MPS-treated Gli1^+^ MMPs (Figure 3C). Pseudotime analysis was performed by Monocle2 to reconstruct cell developmental trajectories. The results show that there were two branches both originating from COP, one oriented towards MALP and the other to OB (Figure 3D). MPS treatment predisposed COP cells towards the MALP branch, and the proportion of cells differentiated into OB was shrank compared with the Gli1^+^ MMPs from Veh injection mice (Figure 3D) which suggested that MPS facilitated Gli1^+^ MMPs undergoing adipogenesis instead of osteogenesis. To confirm this finding, the one-month-old Gli1-CreER^T2^; tdTomato mice were treated with MPS for 5 days and then labeled Gli1^+^ MMPs by Tamoxifen administration. Since the OB cluster of Gli1^+^ cells highly expressed *Osx* (Figure 3B), we then performed immunostaining with OSX antibody and found that not only the number of tdTomato^+^ cells but also the proportion of OSX^+^tdTomato^+^ cells were decreased in the presence of MPS in the chondro-osseous junction. (Figure 3E–H), which was consistent with the single-cell sequencing data. Thus, bioinformatic and genetic experiments revealed that MPS decreased the OB subpopulation of Gli1^+^ MMPs in vivo.

### 2.4. Teriparatide Rescues the Reduction in Bone Mass Caused by MPS Treatment

Teriparatide is a recombinant peptide that is identical to the N-terminal fragment of human parathyroid hormone. Extensive studies indicate that teriparatide promotes bone formation and is widely used in the clinical treatment of osteoporosis [19,20,21]. Accordingly, we asked whether teriparatide rescued the loss of bone mass due to altering the behavior of Gli1^+^ MMPs. To achieve this, we first validated whether teriparatide attenuated the adverse influence of MPS on bone mass. We treated C57BL/6J mice starting from one month of age with MPS, teriparatide (Teri), or both (MPS + Teri) for 28 days. The three-dimensional reconstruction of the distal femur indicated that teriparatide rescued the MPS-decreased bone mass not only in the trabecular bone but also in the cortical bone (Figure 4A,F). Quantification of the μCT data revealed that the mice treated with teriparatide increased the bone volume significantly compared to the normal spongiosa, and also rescued the bone loss caused by MPS (Figure 4B–E). Meanwhile, the mice treated with teriparatide also increased the cortical thickness (Ct. Th) and cortical area (Ct. Ar), without alerting total area (Tt. Ar) in the presence of MPS (Figure 4G–I). The histology of H&E staining and Masson staining confirmed that teriparatide rescued the reduction in bone mass by MPS. The number of trabeculae and the width of cortical bone were both recovered (Figure 4J,K). Thus, our data indicated that teriparatide rescued the MPS-caused bone loss, which was consistent with the previous reports.

### 2.5. Teriparatide Recovers the Proliferation and Differentiation of Gli1^+^ MMPs in the Presence of MPS

Because the teriparatides were demonstrated to increase the proliferation and osteoblast differentiation of the Gli1^+^ MMPs [12], we next investigated whether the teriparatides recovered the proliferation and production of Gli1^+^ MMPs in the presence of MPS. The Gli1-CreER^T2^; tdTomato mice were TAM exposed and then administrated vehicle (Veh), MPS, teriparatide (Teri), or both (MPS + Teri) for five days. Notably, teriparatide attenuated the inhibition of MPS on Gli1^+^ MMPs’ number within the metaphysis (Figure 5A,B). Agreeing with this, the EdU staining also showed that teriparatide relieved the reduction in the Gli1^+^ MMPs’ proliferation caused by MPS (Figure 5C,D). Likewise, teriparatides also enlarged the ratio of the OSX^+^tdTomato^+^ cells, which showed that teriparatide attenuated the MPS-decreased osteogenic differentiation (Figure 5E,F). Consistent with previous studies [22,23], we observed teriparatide promoted angiogenesis even under the condition of MPS by Endomucin immunostaining (Figure 5G,H).

### 2.6. MPS Alters the Glucose Metabolism in Gli1^+^ MMPs

Dexamethasone has been reported to decrease the levels of the plasma membrane-associated glucose transporter GLUT1 [22], and targeting the sodium–glucose cotransporter 2 (SGLT2) ameliorates glucocorticoid-induced osteoporosis [24]. Given the fact that glucose flux is indispensable for MSCs’ proliferation and differentiation [25,26]. Therefore, we hypothesized whether MPS affected the glucose metabolism of Gli1^+^ MMPs as a possible mechanism causing the reduction in cell number and less osteogenesis. Gli1^+^ MMPs were isolated as described before, and then treated with 10 μM MPS or vehicle for three days before Seahorse analysis. MPS treatment notably reduced OCR not only under basal conditions but also in response to the modulators revealing the mitochondrial function (Figure 6A). In particular, MPS decreased basal respiration (Figure 6C), maximal respiration (Figure 6D), ATP production (Figure 6E), and spare respiratory capacity (Figure 6F). Meanwhile, MPS also impaired ECAR (Figure 6B). Next, we extracted proteins from Gli1^+^ MMPs and detected the expression levels of glycolytic enzymes including HK2, PDK1, LDHA, and Glut1 by Western blotting. Our data showed that the expression of Glut1 was much less in response to MPS, without affecting the expression of HK2, PDK1, and LDHA (Figure 6G), indicating that MPS affected the glucose uptake and damaged the glycolysis in Gli1^+^ MMPs.

## 3. Discussion

Synthetic glucocorticoids (GCs) are widely used to cue inflammation and autoimmune diseases; however, their severe side effects on bone health and fracture risk have been extensively reported. The GCs increase the ratio of the receptor activator of nuclear factor-κΒ (RANK)–RANK ligand (RANKL) versus osteoprotegerin, therefore resulting in promoting differentiation and maturation of osteoclasts and enhancing bone resorption [27,28]. The long-term effects of GCs disrupt the osteoclast cytoskeleton leading to decreased osteoclast activity, despite elongating their lifespan [29,30]. On the other side, GCs upregulate adipogenic gene expression, including PPARγ2 [31], C/EBPα, and AP2 [32], causing the preferential differentiation of MSCs to adipocytes instead of osteoblasts, therefore decreasing the number of osteoblasts. Although a large number of studies are focused on the mechanism, the target cell types mediating induced osteoporosis need to be further elucidated.

The Gli1^+^ MMPs, located in the chondro-osseous junction in the long bone, are indispensable for trabecular bone formation and bone fracture healing in adulthood. Our previous finding demonstrates these progenitors possess high heterogeneity, including COP, MALP, OB, and preOB. Based on the trajectory analysis, the subpopulation termed COP sits on top of the hierarchy and differentiates into the other three clusters. Of note, within the osteoblast lineage, COP is considered the progenitor that gives rise to the osteoblast subpopulation of Gli1^+^ MMPs.

In this study, we observe MPS decreases the proliferation and differentiation of Gli1^+^ MMPs, which further reduces bone mass. Importantly, the OB cluster of Gli1^+^ MMPs is dramatically decreased in response to MPS analyzed by scRNA-seq and lineage tracing experiments indicating the OB cluster is a potential cell type mediating MPS-induced osteoporosis. In our previous study, teriparatide expanded the COP subpopulation of Gli1^+^ MMPs, which gave rise to other clusters including OB. Therefore, the rescue effect of teriparatide on GC-induced osteoporosis may function through Gli1^+^ MMPs but in distinct differentiation statuses (Figure 3C). Thus, Gli1^+^ cells are not only indispensable for bone homeostasis but also the targets of GC-induced bone formation defects. However, our study did not demonstrate the specific molecular mechanism of the effect on Gli1^+^ MMPs in vivo experiments. Meanwhile, we did not provide direct evidence that the teriparatide-increased COP cluster functionally rescued the OB cluster in vivo. In the future, we will take advantage of the Dual Cre and Dre recombinases system, specifically labeling the COP subpopulation which co-expresses Gli1 and Aggrecan and demonstrate that these COP cells indeed rescue the OB subpopulation reduced by MPS in vivo.

## 4. Materials and Methods

### 4.1. Animal Model and Treatment

Male C57BL/6J mice, 1 month of age, were obtained from Chengdu Dossy Experimental Animals Co., Ltd. (Chengdu, China). To establish a glucocorticoid-induced model of osteopenia, MPS (25 mg/kg/day) or vehicle was administered by daily intraperitoneal injection to 1-month-old male C57BL/6J mice for 4 weeks. The genotype of Gli1-CreER^T2^; tdTomato mice was obtained by crossing the Gli1-CreER^T2^ (Jackson lab (Bar Harbor, ME, USA), strain #007913) mice with the tdTomato (Jackson lab, strain #007909) mice. The Gli1-CreER^T2^; tdTomato mice were administrated Tamoxifen (TAM) (APExBIO (Houston, TX, USA)) intragastrically at the age of one month for three consecutive days. The dose of the TAM was used as 2 mg/30 g body weight. After TAM administration, Gli1-CreER^T2^; tdTomato mice were injected with MPS or vehicle daily for 5 days. The 5-ethynyl-2′-deoxyuridine (EdU) was dosed at 2 mg/g body weight and injected 4 h before harvest. For the rescue experiments, the dosage of Teriparatide was 0.4 mg per gram of body weight. All the mice were kept in a specific pathogen-free facility at Sichuan University with a 12 h light and 12 h dark cycle in a temperature-regulated room with a standard chow diet. All mice experiments were approved by the Institution Animal Care and Use Committee (IACUC) of Sichuan University of West China Hospital (No. 20220411002).

### 4.2. Micro CT Analysis

After 4 weeks of treatment, femora were collected from mice, fixed in 4% paraformaldehyde overnight, and analyzed by μCT45 (Scanco Medical AG, Brüttisellen, Switzerland). The following scanner settings were used. X-ray tube potential, X-ray intensity, and integration time were set to 55 kVp, 145 μA, and 250 ms, respectively. To quantify trabecular bone parameters, 100 sections below the distal growth plate of the femur have been analyzed. Meanwhile, 50 slices were analyzed in the middle of the femur for cortical bone analysis. The thresholds of trabecular and cortical bone analysis were 300 and 380, respectively. The trabecular BV/TV, trabecular thickness (Tb. th), trabecular number (Tb. N), and trabecular separation (Tb. Sp) were conducted. Cortical bone was analyzed to determine the cortical thickness (Ct. Th), cortical area (Ct. Ar), and total area (Tt. Ar), and 3D images were reconstructed for visualization.

### 4.3. Histological Analysis

For histologic analysis, femurs were harvested from mice and fixed in 4% PFA overnight at room temperature. Femurs were decalcified with 14% EDTA for 2 weeks. Paraffin-embedded and sectioned at 7 μm thickness using a paraffin microtome after decalcification and dehydration. Hematoxylin and eosin (H&E) and Masson staining were performed on paraffin sections. Stained images of the paraffin section were acquired by SLIDEVIEW VS200 (Olympus, Tokyo, Japan).

### 4.4. Bone Dynamic Analysis

MPS- and vehicle-treated 4-week-old mice were labeled with calcein (10 mg/kg, Sigma (St. Louis, MO, USA)) 7 days before harvest and alizarin red (30 mg/kg, Sigma) 2 days before sample collection by intraperitoneal injection. Undecalcified femurs were fixed with 4% PFA and 30% sucrose was used to dehydrate. For microscopic analysis, ten micrometer frozen sections were performed.

### 4.5. Frozen Section and Immunofluorescence Staining

The tibiae were isolated from mice and then fixed in 4% PFA overnight at 4 °C. After 3 days of decalcification in 14% EDTA, the tibiae were immersed in 30% sucrose overnight to dehydrate and then embedded in O.C.T Compound (Tissue-Tek (Torrance, CA, USA)). Sections at a thickness of 10 μm were prepared using a cryostat (Leica (Wetzlar, Germany), CM1950). The section incubated with the primary antibody overnight of Aggrecan (Sigma-Aldrich, AB1031, 1:50), Endomucin (Santa Cruz (Santa Cruz, CA, USA), sc-65495, 1:50), and Osx (Abcam (Cambridge, UK), ab22552, 1:200). The secondary antibody goat anti-rabbit Alexa Flour 488 or goat anti-rat Alexa Flour 488 (Thermo-Fisher Scientific (Waltham, MA, USA), 1:200) was used to incubate for one hour at room temperature. A Click-It EdU Alexa Flour 488 Imaging Kit (Thermo-Fisher Scientific, C10337) was used to detect the expression of EdU. All immunofluorescence staining sections were observed using an Olympus FV3000 confocal microscope.

### 4.6. Preparation of Bone Cell Suspension under the Growth Plate

The distal epiphysis of the femur and proximal epiphysis of the tibia was gently removed, and the metaphyseal region was dissected in FACS buffer (10%FBS in PBS) for approximately 5 mm. The isolated metaphysis was digested with 3 mg/mL collagenase type I (Sigma-Aldrich, C0130) in α-MEM (GIBCO) for 30 min at 37 °C and shaken gently. Cells were filtered through a 70 μm cell strainer and centrifuged to obtain the cell pellet, then re-suspended with FACS buffer to receive the cell suspension for further experiments.

### 4.7. Cell Culture and Seahorse Extracellular Flux Assay

The Gli1-CreER^T2^; tdTomato mice at the age of one month were administered with TAM for 5 days before harvest. Metaphysis mesenchymal progenitors were extracted as described above and amplificated on the plates with 20% FBS (ZETA (New York, NY, USA)) and 1% Pen Strep (Gibco (Waltham, MA, USA)) in α-MEM (Gibco). After one passage, the cells were digested with Accutase (BioLegend (San Diego, CA, USA)) and the PE-positive cells were sorted with BD Arial III. The tdTomato-positive cells were exposed to 10 μM MPS or vehicle for three days before the seahorse extracellular flux assay. Prepared the Seahorse Media and customer Media on the first day. Both the Seahorse Media and the Customer Media contained 5.5 mM glucose, 0.1 mM pyruvate, and 2 mM glutamine. The XF24 plates were coated with 100 μg/mL Poly-D Lysis and stored at 4 °C overnight. On the next day, the treated Gli1^+^ MMPs were seeded with 40,000 cells/well, then changed to Seahorse Media and incubated for one hour. In total, 5 μM Oligomycin, 1 μM Carbonyl cyanide-4 (trifluoromethoxy) phenylhydrazone (FCCP), Rotenone and Antimycin A were added into the built-in injection ports on XF sensor cartridges, respectively. Finally, we ran the mito-stress protocols on the XFe24 Analyzers.

### 4.8. Single-Cell RNA Sequencing and Analysis

According to the manufacturer’s protocol, Chromium Single Cell 3′ Reagent v3.1 kits were used to prepare libraries, and then the libraries were sequenced on a chromium single-cell controller instrument, and 150 bp paired-end reads were generated. Raw sequencing reads were demultiplexed and aligned to the mouse reference genome GRCm38/mm10 using 10× Genomics pipeline cell ranger v3.0 (https://support.10xgenomics.com (accessed on 14 February 2019)) with default parameters, generating a unique molecular identifier count matrix for downstream analysis. Expression matrixes were imported into Seurat v4.3 in R v4.2.2, and doublets and low-quality cells were excluded by limiting the percentage of mitochondrial transcripts smaller than 20%, and the identified gene number in each cell ranged from 200 to 6500. In total, 1916 cells remained after quality control, and the NormalizeData function was used to logarithmically normalize the UMI count matrix of these remaining cells. The Harmony (version 1.2.0, https://github.com/immunogenomics/harmony (accessed on 12 November 2023)) package was used for batch effect correction to avoid batch differences. For dimensionality reduction and visualization, principal component analysis was performed based on 2500 variable features. The top 8 principal components were selected for non-linear dimensional reduction through t-distributed stochastic neighbor embedding (tSNE), and the main clusters were identified using the “FindClusters()” with resolution = 0.25. Conventional markers described in a previous study were used to categorize every cell into a known biological cell type [12]. The data on the vehicle were downloaded from NCBI (GEO: GSE169560).

### 4.9. Western Blotting

Metaphysis mesenchymal progenitors were extracted and cultured as described above. RIPA (Beyotime (Shanghai, China), P0013B) lysis buffer containing a 10% protease and phosphatase inhibitor cocktail was used to isolate total protein from Gli1^+^ MMPs. We separated the protein samples by SDS-PAGE and transferred them into PVDF (Millipore (Burlington, MA, USA), ISEQ00010) membranes. PVDF membranes containing proteins were blocked with 5% BSA for 1 h before incubating with primary antibody for HK2 (Cell Signaling Technology (Danvers, MA, USA), C64G5, 1:1000), LDHA (proteintech (Rosemont, IL, USA), 19987-1-AP, 1:1000), PDK1 (ENZO life science (Farmingdale, NY, USA), ADI-KAP-PK112, 1:1000), Glut1 (Abcam, ab115730, 1:1000) and anti-β-actin (Beyotime, AF5003, 1:1000) for about 16 h at 4 °C. After washing three times with PBST for 5 min, the PVDF membranes were incubated with goat anti-rabbit IgG H&L (HRP) (Abcam, ab66721, 1:5000) for one hour at room temperature and visualized using the Ori Supersensitive ECL Kit (Oriscience (Chengdu, China), PD203).

### 4.10. Quantification and Statistical Analysis

Comparisons between the two groups were made using unpaired Student’s *t*-tests. A one-way analysis of variance (ANOVA) followed by a post hoc Tucky test was used for multiple comparisons. *p*-values less than 0.05 are considered to be statistically significant. At least three independent experiments were used to select all representative images. Error bar refers to standard deviation.

## 5. Conclusions

In vivo experiments have demonstrated that MPS affects the proliferation and osteogenic differentiation of Gli1^+^ MMPs, resulting in reduced bone mass in mice. Single-cell sequencing proved that MPS mainly reduced the OB subpopulation in Gli1^+^ MMPs. Meanwhile, teriparatide can rescue the effects of MPS on bone mass and Gli1^+^ MMPs. Moreover, in vitro experiments have revealed that MPS diminishes the expression of Glut1 in Gli1^+^ cells, affecting the OXPHOS and glycolysis.

## Figures and Tables

**Figure 1 ijms-25-04371-f001:**
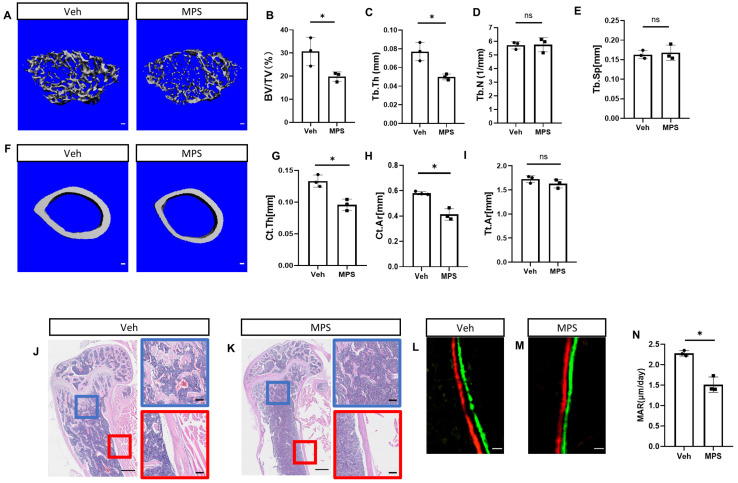
Methylprednisolone can cause osteopenia. (**A**,**F**) C57BL/6J mice were divided into two groups randomly and were treated with methylprednisolone (MPS) at 25 mg/kg/day or vehicle (Veh) by daily intraperitoneal injection for 28 days. (**A**) μCT images of trabecular bone of the distal femur. Scale bar: 100 μm. (**F**) μCT images of cortical bone in the middle of the femur. Scale bar: 100 μm. (**B**–**E**) Quantification of BV/TV, Tb. Th, Tb. N, Tb. Sp by μCT. Error bar: SD. * *p* < 0.05, *n* = 3, Student’s *t*-test. (**G**–**I**) Quantification of Ct. Th, Ct. Ar, Tt. Ar by μCT. Error bar: SD. * *p* < 0.05, *n* = 3, Student’s *t*-test. (**J**,**K**) The H&E staining of longitudinal sections through the distal femur. Scale bar: 500 μm. Blue boxes denote the higher magnification images of trabecular bone. Red boxes denote the higher magnification images of cortical bone. Scale bar: 100 μm. (**L**,**M**) Representative images of double labeling for bone formation on the endosteal surface of the tibia. Green, calcein; red, alizarin red. Scale bar: 10 μm. (**N**) Quantification of the mineral apposition rate (MAR) at the endosteal bone surface from double labeling experiments. Error bar: SD. * *p* < 0.05, *n* = 3, Student’s *t*-test.

**Figure 2 ijms-25-04371-f002:**
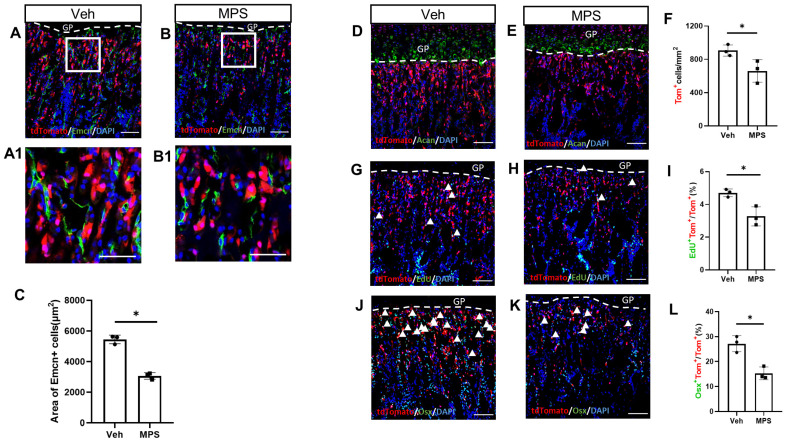
Methylprednisolone decreases the proliferation and osteoblast differentiation of MMPs. (**A**–**K**) Mice with the genotype of Gli1-CreER^T2^; tdTomato were injected with methylprednisolone (MPS) or vehicle (Veh) for five days after TAM treated for three days. (**A**,**B**) Representative confocal images showing direct fluorescence of tdToamto and immunostaining of Endomucin. Boxed areas are shown below at a higher magnification (**A1**,**B1**).Scale bar: 50 μm. The dotted line indicates the boundary between the growth plate and the primary spongiosa. Same below. (**C**) Quantification of the area of the Endomucin positive cells among the 300 μm and spanning the width of the bone flanked by the periosteum under the growth plate. (**D**,**E**) Representative confocal images of Aggrecan. (**G**,**H**,**J**,**K**) Representative confocal images of primary spongiosa below the growth plate in the proximal tibia showing colocalization of tdTomato with EDU (**G**,**H**) or Osx (**J**,**K**). (**F**) Quantification of tdTomato^+^ cells in the primary spongiosa region 300 μm below the growth plate. (**I**,**L**) Quantification of EdU and tdTomato double-positive cells (**I**) or Osx and tdTomato double-positive cells (**L**) among the tdTomato-positive cells in the primary spongiosa. White arrowheads point to the double-positive cells in the primary spongiosa. Scale bar: 100 μm. Error bar: SD. * *p* < 0.05, *n* = 3, Student’s *t*-test.

**Figure 3 ijms-25-04371-f003:**
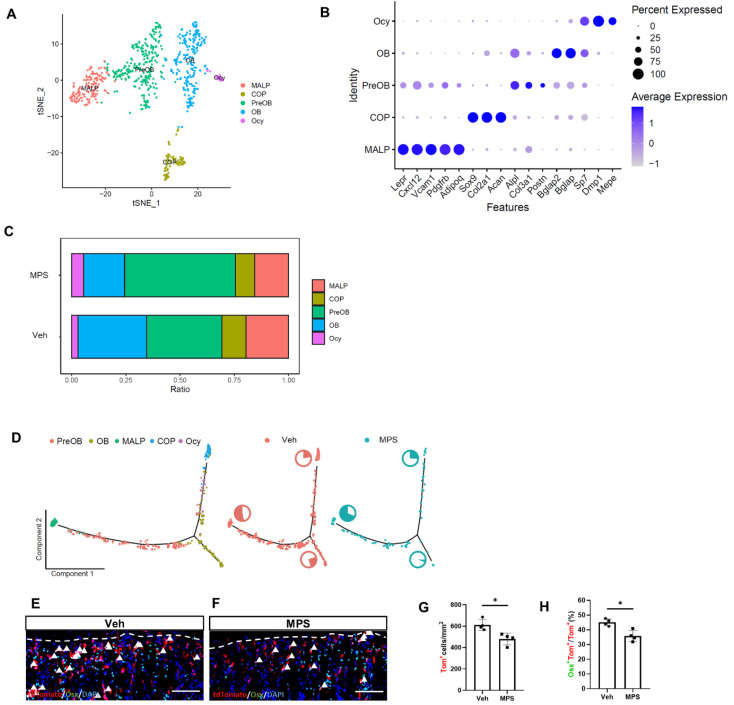
scRNA-seq reveals heterogeneity of MMPs. (**A**) Visualization of clusters from integrated analysis vehicle and methylprednisolone datasets. (**B**) Dotplot of representative marker genes for each cluster. (**C**) Relative abundance of each cluster in MMPs of vehicle- versus methylprednisolone-treated mice. OB, osteoblast; preOB, preosteoblast; COP, chondrocyte-like osteoprogenitor; MALP, marrow adipocyte lineage progenitor; Ocy, osteocyte. (**D**) Trajectory inference of osteogenesis subsets. Left: distribution of each cluster on the branches. Center and Right: distribution of the cells from Veh- or MPS-treated group on the branches; The pie chart next to each branch represents the proportion of the number of cells on that branch to the total number of cells. (**E**,**F**) Mice with the genotype of Gli1-CreER^T2^; tdTomato were injected with methylprednisolone (MPS) or vehicle (Veh) for five days, and then treated with TAM for three days. Representative confocal images display the tdTomato and immunostaining of Osx in the primary spongiosa region of the proximal tibia. (**G**) Quantification of tdTomato^+^ cells in the primary spongiosa region 300 μm below the growth plate and extends the width of the bone flanked by the periosteum. (**H**) Quantification of double-positive cells in the primary spongiosa as presented in (**E**,**F**). Arrowheads denote double-positive cells in the primary spongiosa. Dotted lines denote the boundary of the growth plate. Scale bar: 100 μm. Error bar: SD. * *p* < 0.05, *n* = 4, Student’s *t*-test.

**Figure 4 ijms-25-04371-f004:**
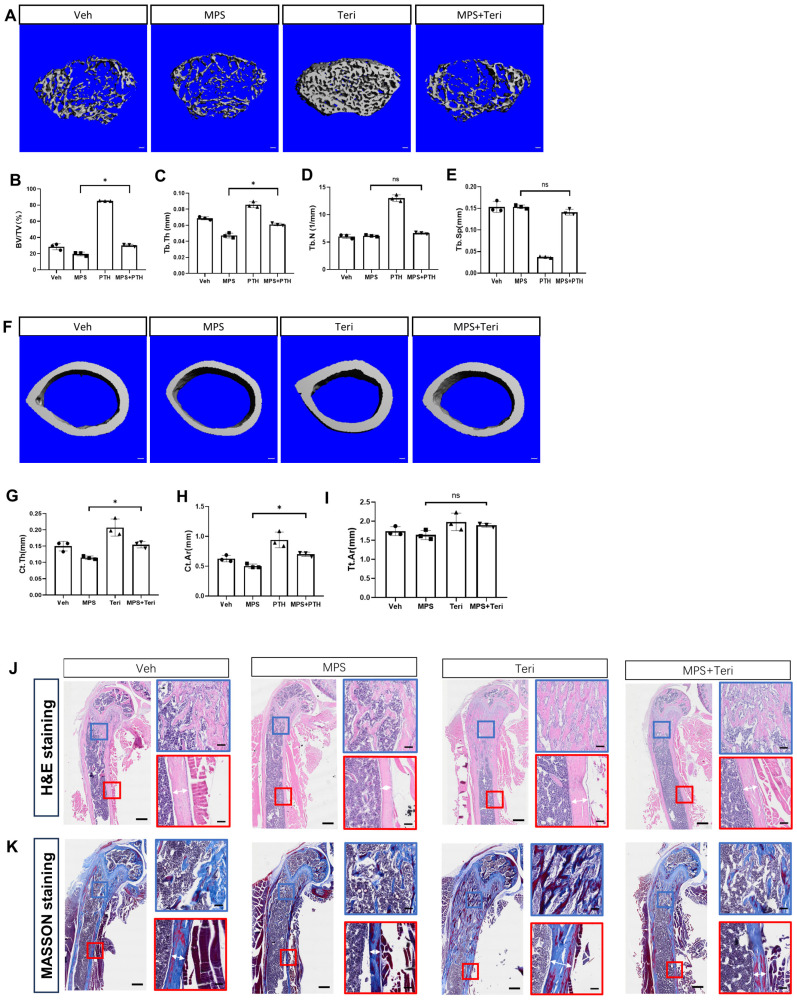
Teriparatide can rescue the bone mass reduction caused by methylprednisolone. (**A**–**I**) Male wild-type mice were divided equally into four groups. They were injected with methylprednisolone (MPS), teriparatide (Teri), methylprednisolone, and teriparatide (MPS + Teri), or vehicle (Veh) at the age of one month for consecutive 28 days. (**A**) μCT images of cancellous bone of the distal femur. Scale bar: 100 μm. (**F**) μCT images of cortical bone in the middle of the femur. Scale bar: 100 μm. (**B**–**E**) Quantification of BV/TV, Tb. Th, Tb. N, Tb. Sp by μCT. (**G**–**I**) Quantification of Ct. Th, Ct. Ar, Tt. Ar by μCT. (**J**,**K**) The H&E staining and MASSON staining of longitudinal sections through the distal femur. Scale bar: 500 μm. Blue and red boxes denote the higher magnification images of trabecular bone and cortical bone, respectively. Scale bar: 100 μm. Double-headed arrow shows the width of the cortical bone. Error bar: SD. * *p* < 0.05, *n* = 3, one-way ANOVA.

**Figure 5 ijms-25-04371-f005:**
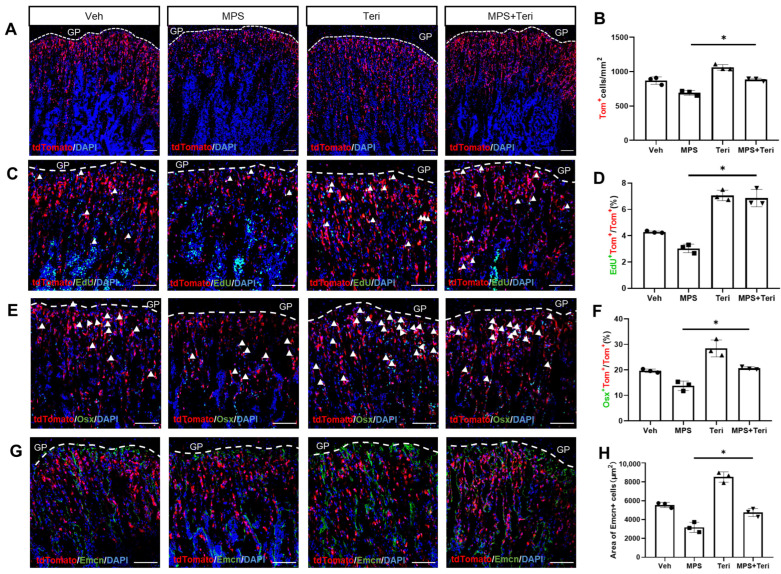
Teriparatide can rescue the proliferation and osteogenic differentiation of MMPs by methylprednisolone. (**A**,**C**,**E**,**G**) The genotype of Gli1-CreER^T2^; tdTomato mice were injected with methylprednisolone (MPS), teriparatide (Teri), methylprednisolone, and teriparatide (MPS + Teri), or vehicle (Veh), respectively for five days at the age of one month after Tam induced for three days. (**A**) Confocal images of tdTomato direct fluorescence in different groups. Scale bar: 100 μm. Dotted lines denote the boundary of the growth plate. Same below. (**B**) Quantification of tdTomato^+^ cells in the primary spongiosa region 300 μm below the growth plate. (**C**,**E**) Representative confocal images indicate direct fluorescence of tdToamto and immunostaining of EdU and Osx, respectively. Scale bar: 100 μm. Arrowheads denote double-positive cells in the primary spongiosa. (**D**,**F**) The statistical analysis of EdU^+^tdTomato^+^ cells (**D**) and Osx^+^tdTomato^+^ cells (**F**) over the tdTomato^+^ cells in the primary spongiosa of the proximal tibia, extending 300 μm from the growth plates and across the width of the bone. (**G**) Representative confocal images of immunofluorescence staining of Endomucin. (**H**) Quantification of the area of the Endomucin positive cells among 300 μm in the metaphysis region below the growth plate. Error bar: SD. * *p* < 0.05, *n* = 3, one-way ANOVA.

**Figure 6 ijms-25-04371-f006:**
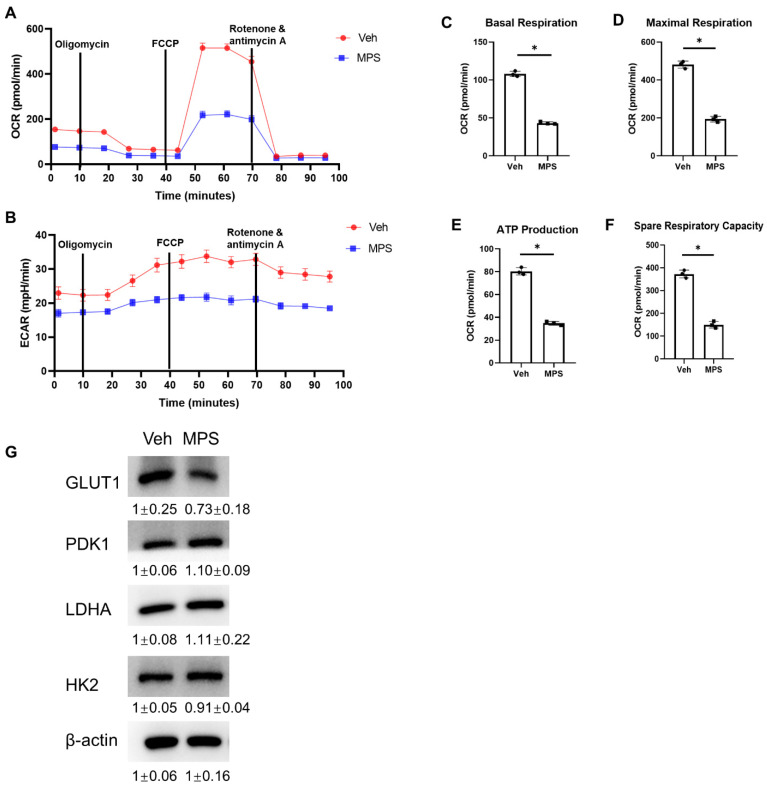
Methylprednisolone affects the glucose metabolism of Gli1^+^ progenitors. (**A**) OCR: oxygen consumption rate; (**B**) ECAR: extracellular acidification rate; (**C**–**F**) basal respiration (**C**), maximal respiration (**D**), ATP production (**E**), and spare respiratory capacity (**F**). (**G**) Western blot was performed to detect the protein expression levels of Glut1, PDK1, LDHA, HK2, and β-actin in Gli1-positive cells exposed to vehicle and MPS, respectively. Numbers below the strip denote the grey values (mean ± SD) analyzed by ImageJ version2.1.0. Error bar: SD. * *p* < 0.05, *n* = 3, Student’s *t*-test.

## Data Availability

The original data of single-cell sequencing under the treatment of MPS and Veh are openly available in NCBI, the GEO numbers are GSE248553 and GSE169560, respectively. The other raw data supporting the conclusions of this article will be made available by the authors on request.

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
