# Peer review of "Gli1+ Progenitors Mediate Glucocorticoid-Induced Osteoporosis In Vivo"

_ijms, 2024, doi:10.3390/ijms25084371_

Round 1

Reviewer 1 Report

Comments and Suggestions for Authors

- if the authors consider to expand the introduction section, they should include the therapeutic limitations by discussing the clinical impact of the adverse effects discussed

- summarize the mechanisms GIO that have been so far demonstrated

- in the introduction, the reason of choosing teriparatide for this study should be discussed in the context of GIO and GC effects

- I recommend that the introduction section will end in the aim or hypothesis of the study

- for more scientific clarity, authors should choose between: Celsius, ℃, room temperature

- is there any reason for why specific C57BL/6J mice were chosen?

- briefly justify the chosen dosages and treatment durations for MPS, TAM, and EdU

- IJMS is a promotor of science reproducibility, for this reason, authors should provide protocols, including concentrations, incubation times, and specific conditions for each step of the assays and analyses (for tissue samples too)

- what were the treatment durations and schedules for all groups?

- describe how the statistical tests of were applied for assumptions of normality and homogeneity

- readers might be interested in explaining what are the hypothesized molecular pathways through which teriparatide acts on Gli1+ MMPs?

- I thoroughly encourage authors to include a conclusion section for their article.

- references are adequate

- the title is relevant, short and concise

- english level is advanced

Reviewer 2 Report

Comments and Suggestions for Authors

The manuscript titled "Gli1+ Progenitors Mediate Glucocorticoid-Induced Osteoporosis in vivo" presents commendable research efforts by the authors. This engaging submission investigates the impact of daily administration of methylprednisolone (MPS), a synthetic glucocorticosteroid (GC), on the osteogenic differentiation and proliferation of Gli1+ metaphyseal mesenchymal progenitors (MMPs). Additionally, the authors explore whether teriparatide-induced chondrocyte-like osteoprogenitors (COP) contribute to the osteoblasts (OB) subpopulation of Gli1+ MMPs and consequently increase bone mass.

The manuscript is commendably clear and readable, with a well-structured introduction, precise materials and methods section, conclusive results, concise discussion, and up-to-date references. The inclusion of high-quality illustrative figures significantly enhances the overall comprehensibility of the manuscript.

However, before considering this manuscript suitable for publication, a pivotal aspect requires further clarification. It is recommended that the authors clearly outline the study limitations in the discussion section. Additionally, a firm conclusion providing perspective on further research avenues would be beneficial. Finally, specifying the Data Availability Statement is essential.

In conclusion, while this manuscript holds promise, addressing the aforementioned points is crucial to enhancing its impact on the field and improving its suitability for publication.

Round 2

Reviewer 1 Report

Comments and Suggestions for Authors

Authors performed the changes requested.

Reviewer 2 Report

Comments and Suggestions for Authors

Well done.